# Sound Quality Estimation of Electric Vehicles Based on GA-BP Artificial Neural Networks

**Kun Qian** [1],*, **Zhichao Hou** [1] **and Dengke Sun** [2]

1    State Key Laboratory of Automotive Safety and Energy, Tsinghua University, Beijing 100084, China;
      houzc@tsinghua.edu.cn
2    School of Mechanical Electronic & Information Engineering,
      China University of Mining & Technology-Beijing, Beijing 100083, China; sundengke_nvh@163.com
*    Correspondence: qiankun_nvh@163.com; Tel.: +86-010-6279-7088

**Abstract:** The sound quality (SQ) and sound perception assessments of electric vehicles (EVs) clearly differ from those of conventional internal combustion engine vehicles (ICEVs). Therefore, it is essential to describe and evaluate the SQ of EVs. To evaluate the SQ in EVs, it is necessary to organize evaluators for conducting subjective jury tests, which are time-consuming and labor-intensive. In addition, the evaluation results are subject to the evaluators themselves and other external interferences. With the advancement of machine learning and artificial neural networks (ANNs), this problem can be well solved. This paper outlines a model for SQ estimation in EVs based on a genetic algorithm-optimized back propagation artificial neural network (GA-BP ANN). Moreover, the correlation between the physical-psychoacoustical parameters and the subjective SQ estimations obtained from the jury tests was investigated in this study. It was found that the GA-BP ANN SQ model has many advantages in comparison with the multiple linear regression (MLR) model in terms of precision and generalization. In addition, this method is ready to be applied for rapidly evaluating the SQ in EVs without jury tests, and it can also be of high significance in dealing with the acoustical designs and improvements of EVs in the future.

**Keywords:** electric vehicles; noise; sound quality; artificial neural network; genetic algorithm

## 1. Introduction

Nowadays, electric vehicles (EVs), which are driven by electric motors, are becoming very popular because they do not emit pollutants to the environment. Owing to the lack of the internal combustion engine noise and its masking effect in EVs, the noise level of EVs is much lower than that of conventional internal combustion engine vehicles (ICEVs), as shown in Figure 1a. In addition, the aerodynamic noise and road noise contribution to the interior noise in EVs is significant in comparison with ICEVs. Moreover, on the basis of the new noise components of electric motors, the sound characteristics are different, as shown in Figure 1b. The electromagnetic noise, which is unique to EVs, makes the subjective perception very irritating [1,2]. For example, drivers and passengers are often plagued by "howling" and "buzzing" sounds under certain EV operating conditions [3,4]. This largely contributes to the fact that the SQ in EVs has become a major concern for their drivers and passengers [5]. Moreover, the experience in estimating noise in ICEVs cannot be simply used with EVs [6]. Therefore, a new SQ evaluation method is needed for EVs.

At present, the subjective SQ evaluation in EVs is carried out by organizing jury tests, where listening tests are conducted, and each sound sample is subjectively scored and evaluated [7,8]. However, listening tests are not only time-consuming, but they are also influenced by external interferences [9]. If a universally applicable intelligent evaluation model can be established, the subjective

perception of SQ can be directly obtained from the objective parameter of the sound signals in the case of similar events, which can greatly save manpower and material resources. Therefore, on the basis of the physiological and psychological characteristics of auditory systems, it is necessary to study the whole process from the subjective evaluation of sound signal detection and noise processing to modeling so as to realize a rapid and accurate evaluation of the intelligent SQ for EVs.

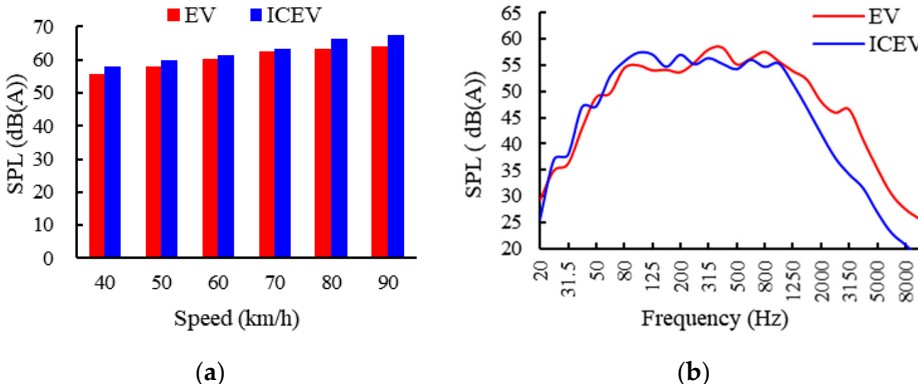

**Figure 1.** Comparison of the interior noise between electric vehicles (EVs) and internal combustion engine vehicles (ICEVs): (**a**) comparison of the interior noise level between EVs and ICEVs at different constant speeds; (**b**) comparison of the noise spectrum in EVs and ICEVs at a certain constant speed. SPL, sound pressure level.

Most of the existing SQ estimation models were constructed using the multiple linear regression (MLR) method [10–14]. In [10], a mathematical model of the preference and annoyance between the objective psychoacoustic parameters and the subjective evaluation was established using the MLR method. Moreover, using correlation and regression analyses, a subjective annoyance model of EVs, which can describe the correlation between the subjective evaluation and the objective parameters, was established [11]. In [12], an SQ subjective evaluation of hybrid electric vehicles was quantified based on the tone, roughness, and sharpness, and a linear regression model of the sample was established. In addition, in [13], through curve estimation and linear regression analysis, an SQ evaluation model of EVs was proposed, where the modeling error between the subjective perception value and the objective parameters was less, and the subjective perception was mainly affected by the loudness, sharpness, and A-weighted sound pressure level (SPL). In [14], a sound index for estimating annoyance using independent variables was developed to evaluate the SQ in trains using statistical analyses. However, the existing MLR model of the SQ cannot directly map the complex nonlinear relationship between the subjective perception and the objective evaluations, and its evaluation accuracy is also relatively low. This is because the relationship between the human subjective perception and the objective parameters is nonlinear and has strong randomness. Generally, there are many factors that lead to differences in the subjective perception of SQ, and these factors often have a nonlinear relationship with subjective perception. Therefore, it is necessary to establish a more accurate nonlinear model.

With the advancement of machine learning and artificial networks, this problem can be well solved. The artificial neural network (ANN) is known as a broadly used approach in vehicle engineering and acoustical engineering, and the relationship between the inputs and outputs in ANNs is nonlinear. Meanwhile, it is possible to design an SQ estimation model by utilizing an ANN to simulate the subjective assessment model in an approximation process of the subjective perception. Many researchers have applied ANNs to SQ evaluation, and they have achieved reliable results [15–20]. Steinbach and Altinsoy [15] used ANNs to predict the annoyance evaluations of the pass-by noise of EVs and found that most of the parameters have no linear influence on the annoyance perception. Gao et al. [16] developed an SQ prediction model of heavy commercial vehicles based on the back propagation artificial neural network (BP-ANN), and their proposed model showed high precision and good generalization. Moreover, Ma et al. [17] established an SQ evaluation model based on the BP-ANN to

evaluate and predict the abnormal noise of permanent magnet synchronous motors in the steady-state working condition of EVs, and Ma et al. [18] evaluated the SQ of EVs under steady-state conditions and established an SQ valuation model based on an ANN. However, the evaluation of the SQ under unsteady conditions needs to be further studied. Zhang et al. [19] proposed a method for predicting the SQ of vehicle interior noise using a BP-ANN that is based on particle swarm optimization (PSO). In addition, Ma et al. [20] established an SQ evaluation model of hub-driven motors based on a BP-ANN, and they also proposed the limitations of the BP-ANN model. In order to improve the prediction accuracy, the structural parameters of BP-ANNs need to be optimized, but this process consumes a lot of time. Generally, it is feasible to describe the subjective perception using objective measurable physical parameters and psychoacoustic parameters. Intelligent algorithms and artificial neural network technologies can be used to complete the comprehensive evaluation of the SQ of EVs instead of the subjective evaluation of humans, where an objective quantitative model for the subjective evaluation of SQ in EVs can be established based on the physiological and psychological characteristics of auditory systems.

In this paper, an SQ estimation model was designed to assess the SQ in EVs by means of the GA-BP ANN. The paper is organized as follows. Section 2 describes the interior sound measurement of EVs, the subjective jury test, and the subjective and objective evaluation of the SQ. In Section 3, the MLR and the GA-BP ANN models of the SQ in EVs are established, respectively, according to the subjective and objective evaluation results. Section 4 discusses the advantages of the GA-BP ANN model in detail by comparing it with the MLR model, and the final section presents the conclusion of the research.

## 2. Measurement and Test

### 2.1. Interior Sound Measurement of EVs

Six various kinds of EVs were used for the test, and the interior noises of these vehicles were collected through the digital artificial head SQuadriga HMS provided by the German company HEAD Acoustics. Binaural headsets-SQuadriga were placed on the front passenger seat, and real binaural sound signals were collected, as shown in Figure 2. The method for measuring the interior noise of vehicles is based on the standard ISO 5128 (acoustics-measurement of noise inside motor vehicles) system. The interior noise test was carried out at the China First Automotive Works (FAW) automobile proving ground. The test road was a straight first-class road with high grade paving asphalt. During the entire test, the air conditioning and windows were closed. The tester and driver should keep quiet. The working conditions of the testing samples were elaborately considered, and they were collected every 10 km/h at a stable speed from 20 km/h to 120 km/h. According to the wind direction, the sound collection for each working condition was performed at least twice. The length of each test sample was set as 30 s. A total of 91 samples with interior sound were collected. Each sound sample was intercepted for 5 s for the subsequent subjective evaluation jury test and objective evaluation analysis.

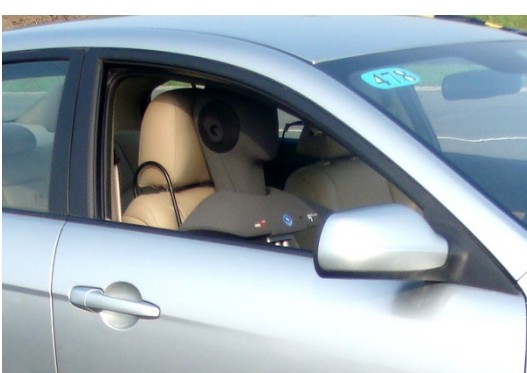

**Figure 2.** Measurement position of the artificial head.

*2.2. SQ Subjective Evaluation*

2.2.1. Listening Jury Test

In order to obtain the subjective score value of each sound sample, it is necessary to organize the evaluation personnel to conduct a listening jury test. In total, 32 participants (8 females and 24 males) with ages ranging from 24 to 52 were recruited in the jury. These participants included drivers, engineers, and acoustic experts who were engaged in the automotive vibration and noise control industry for many years. In order to ensure that the external environment did not interfere with the subjective evaluation tests, the tests were carried out in a semi-anechoic SQ evaluation chamber, as shown in Figure 3. The HEAD SQuare evaluation system was used in this jury test, where it can quickly create simple or complex listening tests as well as customize, save, and repeat the listening tests according to various listening test scenarios.

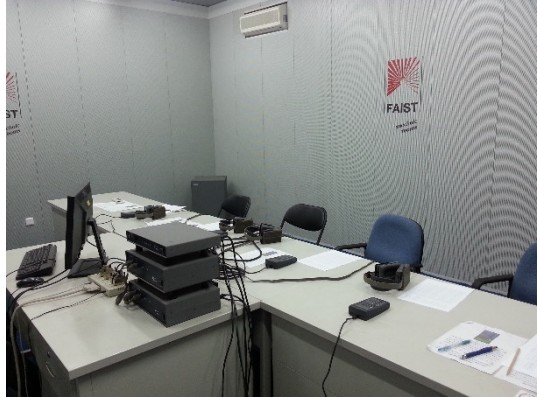

**Figure 3.** Subjective sound quality (SQ) evaluation anechoic chamber.

There are often two main methods for the subjective evaluation of SQ; that is, the grade evaluation method and the paired comparison method. The test in this study employed the grade evaluation method for the subjective listening test and the paired comparison method for training the listening samples before the jury test. On the basis of the sensory pleasantness of the sound, as shown in Figure 4, the scores of the subjective sound perception tests were gauged from point 1 to point 11. Before the formal listening test, all of the samples were randomly scrambled so that the evaluators could make more accurate scores. During the tests, the evaluators listened to different sound samples and scored them based on their own subjective sensations.

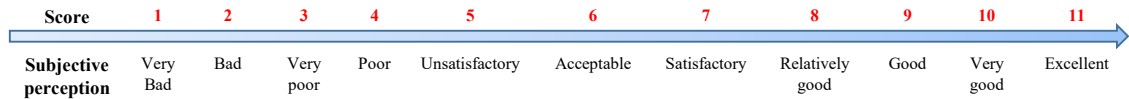

**Figure 4.** Subjective evaluation metric of the SQ.

2.2.2. Evaluation Data Verification

The evaluation data of the evaluators need to be accurately verified, as they are affected by many factors, which may lead to inaccurate evaluation results. Thus, all the obtained subjective evaluation data were validated, and the average correlation coefficient was used as standard to eliminate the results of the unstable evaluators. A correlation analysis was conducted for each evaluator to study the correlation degree of each evaluator's subjective evaluation data. In the grade evaluation method, the Spearman correlation coefficient is usually adopted as the data test metric, as the subjective evaluation result is similar to the variables rank. In the SPSS software, the Spearman coefficients of each evaluator are calculated, and then the correlation coefficients of each evaluator are arithmetically averaged to calculate the average correlation coefficient, as shown in Table 1. It was revealed that the correlation coefficients

of the evaluators of Nos. 10, 21, and 26 were comparatively low, less than 0.7, and the correlation coefficients of the other evaluators were over 0.7. In statistics, a Spearman correlation coefficient below 0.7 signifies a weak correlation. Therefore, the evaluation data of the evaluators of Nos. 10, 21, and 26 were removed.

**Table 1.** Results of the correlation coefficients of the evaluators.

| No. | Correlation | No. | Correlation | No. | Correlation | No. | Correlation |
|-----|-------------|-----|-------------|-----|-------------|-----|-------------|
| 1 | 0.819 | 9 | 0.806 | 17 | 0.827 | 25 | 0.818 |
| 2 | 0.796 | 10 | 0.538 | 18 | 0.818 | 26 | 0.671 |
| 3 | 0.821 | 11 | 0.843 | 19 | 0.821 | 27 | 0.824 |
| 4 | 0.796 | 12 | 0.832 | 20 | 0.834 | 28 | 0.879 |
| 5 | 0.812 | 13 | 0.740 | 21 | 0.614 | 29 | 0.788 |
| 6 | 0.723 | 14 | 0.826 | 22 | 0.803 | 30 | 0.766 |
| 7 | 0.869 | 15 | 0.756 | 23 | 0.798 | 31 | 0.829 |
| 8 | 0.816 | 16 | 0.829 | 24 | 0.853 | 32 | 0.740 |

After removing the three results with the low correlation coefficients, the remaining evaluation results should be clustered based on their individual properties to ensure the reliability of the evaluation results. This classification was performed by an approach of multivariate statistical analysis called the clustering analysis. In this study, the K-means clustering approach was used to analyze the remaining evaluation data, and the results are listed in Table 2. The number of evaluators in the first cluster is too small to represent the results of most of the evaluators. Therefore, the scores of the 25 evaluators in the second cluster were selected to calculate the subjective score of the SQ of each sound sample. The subjective scores for each sound sample were obtained by averaging the scores of the 25 evaluators.

**Table 2.** Clustering results.

| Cluster | Amount |
|---------|--------|
| 1 | 4 |
| 2 | 25 |

*2.3. SQ Objective Evaluation*

The psychoacoustics parameters comprehensively consider the human psychological response mechanism and the acoustic perception characteristics, reflecting the difference in the subjective feelings caused by different sound signals. After obtaining the subjective SQ scores of each sound sample, it is necessary to calculate the psychoacoustics parameters of each sound sample so as to enable a subsequent analysis of the correlation between the subjective and objective evaluation results. By referring to the characteristics and applicability of the existing objective psychoacoustic parameters, the loudness, sharpness, roughness, fluctuation strength, tonality, and articulation index were selected. Meanwhile, the A-weighted SPL was considered. These seven objective parameters can basically represent the psychological characteristics of the sound samples.

As a commonly used noise evaluation index, the A-weighted sound pressure level (SPL) refers to the sound pressure level measured by the A-weighted network. Owing to the large number of sound samples, the noise inside an electric car was taken as an example for analysis. Figure 5 shows that the interior noise SPL gradually increased with the vehicle's speed. Usually, the SPL of the right ear is higher than that of the left ear, as the measuring point of the right ear is close to the outside of the vehicle body, which had more noise.

Loudness is a psychoacoustic parameter reflecting the human ear's perception of sound intensity based on its masking characteristics. The change in the interior noise loudness with the vehicle's speed is shown in Figure 6, where the loudness increased with the vehicle's speed.

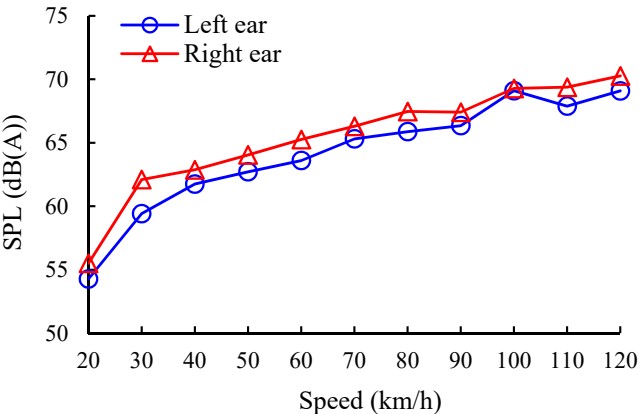

**Figure 5.** Interior noise sound pressure level (SPL) in the EV as a function of the vehicle's speed.

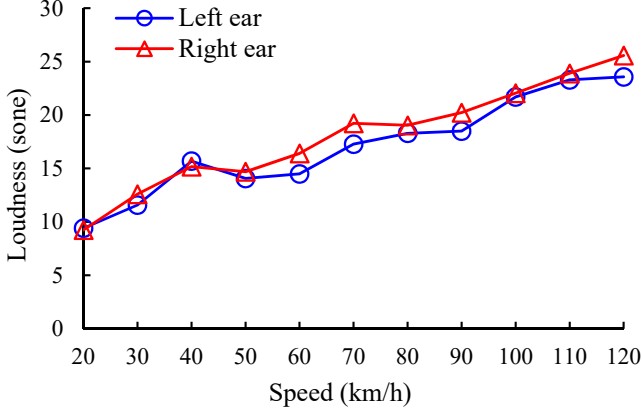

**Figure 6.** Interior noise loudness in the EV as a function of the vehicle's speed.

The sharpness is an objective psychoacoustic parameter that reflects the high-frequency components of sound signals. The change in the interior noise sharpness with the vehicle's speed is shown in Figure 7. At low speeds, the high-frequency motor noise dominates, and the sharpness also increases. As the vehicle speed increases, the motor noise is masked, and the high-frequency components of the wind noise and tire noise dominate.

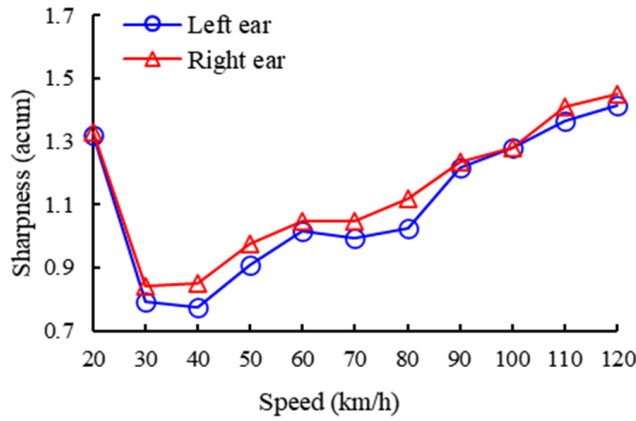

**Figure 7.** Interior noise sharpness in the EV as a function of the vehicle's speed.

The roughness indicates the high-frequency variation of sound signals, while the fluctuation strength (FS) indicates the low-frequency frequency variation of sound signals. Figure 8 shows that the

interior noise roughness gradually increased with the vehicle's speed, and Figure 9 shows that there was no obvious variation trend for the change in the interior noise FS with speed.

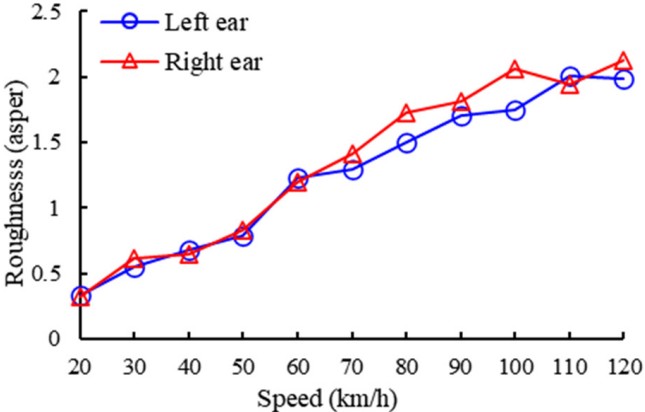

**Figure 8.** Interior noise roughness in the EV as a function of the vehicle's speed.

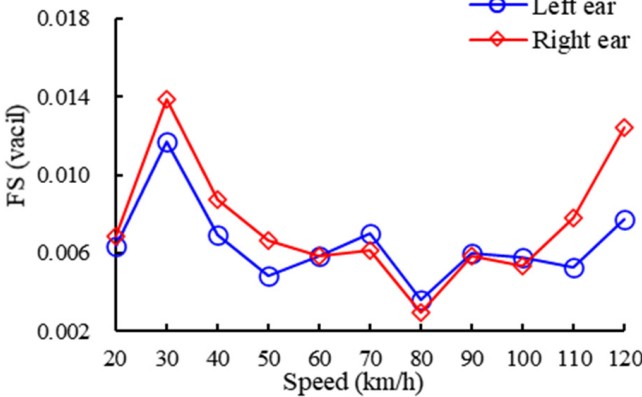

**Figure 9.** Interior noise fluctuation strength (FS) in the EV as a function of the vehicle's speed.

The tonality is a psychoacoustic parameter used to measure the proportion of pure sound components in a spectrum of sound signals. Figure 10 shows that there was no obvious variation trend for the change in the interior noise tonality with speed.

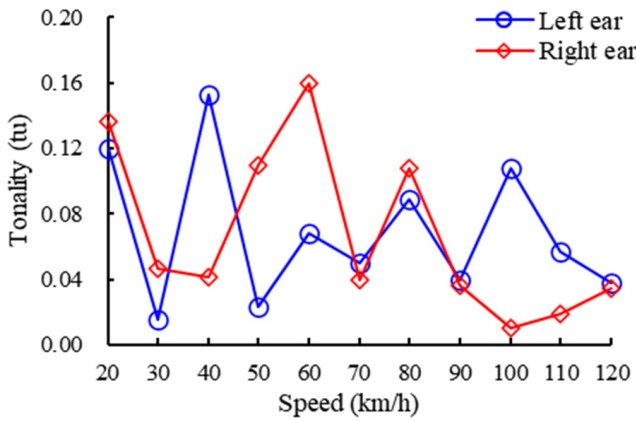

**Figure 10.** Interior noise tonality in the EV as a function of the vehicle's speed.

The articulation index (AI) is a psychoacoustic parameter that reflects the change in the clarity of speech owing to noise in vehicles. Figure 11 shows that the interior noise AI gradually decreased with

the vehicle's speed, which is because the AI is related to the noise level. The greater the noise level, the lower the AI.

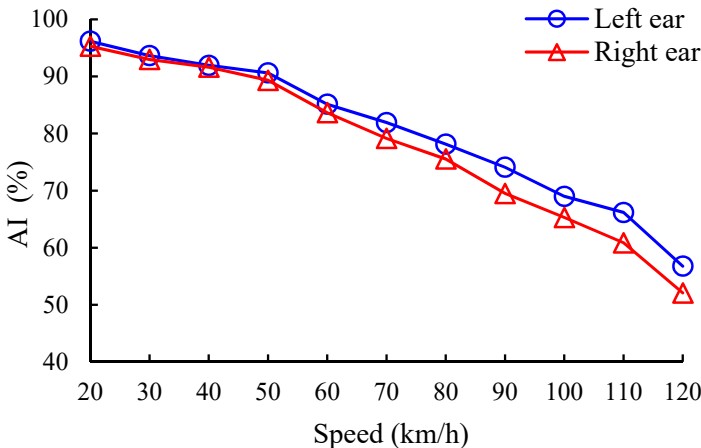

**Figure 11.** Interior noise articulation index (AI) in the EV as a function of the vehicle's speed.

## 3. SQ Evaluation Model of EVs

### 3.1. Multiple Linear Regression Model

Most of the prediction models of SQ are based on theories of multiple linear regression (MLR). Before the MLR analysis, a correlation analysis between the subjective SQ evaluation results and the objective physical-psychoacoustical parameters should be carried out so as to better study the relationship between the subjective SQ evaluation scores and the objective evaluation parameters to determine which objective evaluation parameters have a significant influence on the subjective SQ evaluation of EVs. According to the subjective and objective evaluation results, the SPSS software was used to analyze the relationship between the objective physical-psychoacoustical evaluation parameters and the subjective evaluation scores.

Table 3 shows the correlation coefficient between the objective evaluation parameters and the subjective evaluation scores. It can be seen from the table that the correlation coefficients of the loudness, AI, SPL, roughness, sharpness, and FS are all above 0.7, while the correlation coefficient of the tonality is much lower. Thus, the independent variables were the loudness, AI, SPL, roughness, sharpness, and FS, which had a better correlation with the subjective scores, and the dependent variable was the subjective evaluation score of the SQ. Through the MLR analysis of the subjective evaluation scores and the objective parameters, an objective quantitative SQ model for EVs can be established.

**Table 3.** Correlation coefficients between the sound quality (SQ) scores and the physical-psychoacoustical parameters. SPL, sound pressure level; AI, articulation index; FS, fluctuation strength.

|  | SPL | Loudness | Sharpness | Roughness | AI | FS | Tonality |
|---|---|---|---|---|---|---|---|
| Correlation coefficient | −0.908 ** | −0.928 ** | −0.821 ** | −0.879 ** | 0.917 ** | −0.794 ** | 0.272 * |
| Two-tailed test | 0.000 | 0.000 | 0.000 | 0.000 | 0.000 | 0.000 | 0.017 |

** indicates significant correlation at the 0.01 level (bilateral); * indicates significant correlation at the 0.05 level (bilateral).

After the MLR analysis, there were three models, as shown in Table 4. All the regression coefficients of the models were significant, as the accompanied probability sig values corresponding to the t-statistic of the three models were less than 0.01. This indicates that the three models are reasonable. Moreover, it seems that model 3 is the most suitable model, as the value of the determination coefficient $R^2$ is larger than that of any other model. It also contains a constant term, the loudness, sharpness, and FS variables. In addition, this model had higher correlation and determination coefficients. Consequently,

it was selected as a prediction model. On the basis of the unstandardized coefficient in Table 4, the MLR model of the SQ can be described as

$$SQ = 11.575 - 0.078 \times L - 1.023 \times S - 20.92 \times F, \tag{1}$$

where $SQ$ is the subjective sound quality score, $L$ is the loudness, $S$ is the sharpness, and $F$ is the fluctuation strength. It can be seen from Equation (1) that the loudness, sharpness, and FS are negatively correlated with the subjective evaluation score of the SQ. As seen from Table 4, the absolute loudness value of the standard coefficient 0.517 is greater than that of the others, which indicates that the loudness contributes more to the subjective evaluation score of the SQ.

**Table 4.** Results of the regression analysis.

| Model | Component | Determination Coefficient $R^2$ | Unstandardized Coefficients | | Standard Coefficient | t | Sig |
|---|---|---|---|---|---|---|---|
| | | | B | Standard Error | Beta | | |
| 1 | Constant | 0.876 | 10.277 | 0.111 | | 92.826 | 0 |
| | Loudness | | −0.142 | 0.006 | −0.937 | −23.02 | 0 |
| 2 | Constant | 0.905 | 10.761 | 0.139 | | 77.147 | 0 |
| | Loudness | | −0.116 | 0.008 | −0.766 | −15.23 | 0 |
| | Sharpness | | −0.760 | 0.157 | −0.243 | −4.830 | 0 |
| 3 | Constant | 0.918 | 11.575 | 0.259 | | 44.703 | 0 |
| | Loudness | | −0.078 | 0.013 | −0.517 | −6.231 | 0 |
| | Sharpness | | −1.023 | 0.163 | −0.327 | −6.285 | 0 |
| | FS | | −20.92 | 5.774 | −0.232 | −3.625 | 0.001 |

### 3.2. GA-BP ANN Model

#### 3.2.1. GA-BP ANN

Given the complicated nonlinear correlation of the objective parameters and subjective perception, the artificial neural network (ANN) technique can be used to identify these relationships between the objective parameters and the subjective SQ perception. The back propagation artificial neural network (BP-ANN) can theoretically approximate any function with arbitrary precision to achieve complex nonlinear mapping, which can better solve complex nonlinear problems that are difficult to model with mathematical formulas [21]. Although the BP-ANN has many advantages and is widely used, it still has many shortcomings and defects [22–24]. For example, the robustness is poor, and the network performance is very sensitive to the initial weight setting of the network. The BP-ANN can also easily fall into the local minimum, so it cannot obtain the global optimal solution [25,26].

The genetic algorithm (GA) is a kind of global optimization algorithm established by simulating biological genetic and evolution processes with good global characteristics [27]. It can be used to optimize the BP-ANN, make it have self-evolution, and improve its adaptive ability so as to construct a BP-ANN with a global search ability [28].

Therefore, this study used the GA to optimize the weights and thresholds of the BP-ANN and then build a GA-BP ANN. It also used the global search capability of the GA algorithm to find the weight and threshold corresponding to the initial suboptimal solution of the network, which were then used as the initial weight and threshold of the network. Afterwards, the BP algorithm was used to train the network, and it could well prevent the network from falling into the local minimum, improved the convergence speed, and made the network get rid of its dependence on the initial value. The specific operation flow chart of the GA-BP ANN is shown Figure 12, and the specific steps of the algorithm are as follows: (1) determining the initial structure and parameters of the GA-BP ANN; (2) determining the GA operation, setting the parameters, and encoding; (4) using the obtained optimization weight as the initial network weight and then using the BP algorithm to train the network; and (5) outputting the final result.

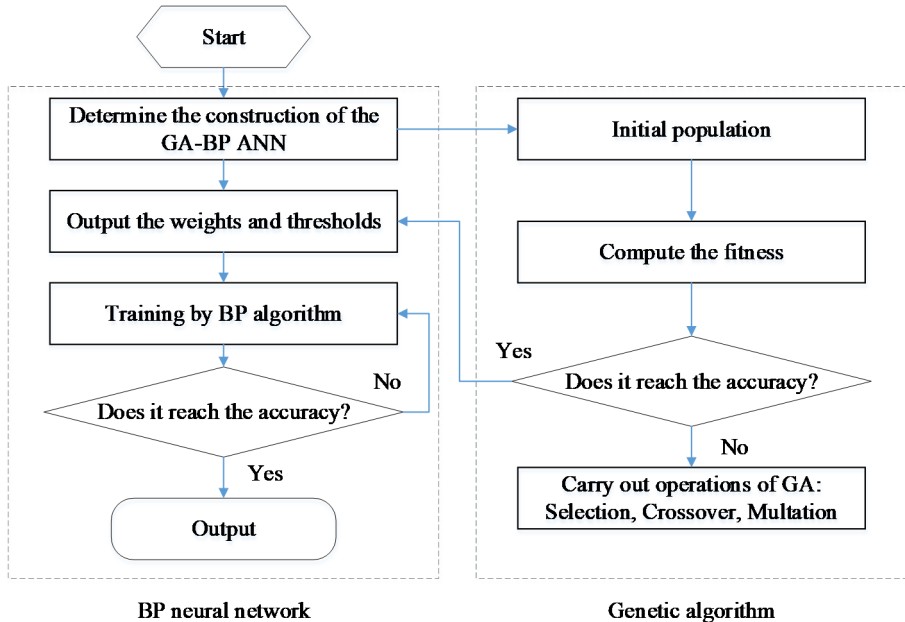

**Figure 12.** The flow diagram of genetic algorithm back propagation artificial neural network (GA-BP ANN).

### 3.2.2. Construction of the GA-BP ANN

The SQ evaluation model of the EVs was constructed using a three-layer neural network of the input layer, output layer, and single hidden layer. The network input layer had seven nodes, which corresponded to the physical-psychoacoustical parameters. There was only one node in the network output layer, and it corresponded to the subjective evaluation scores of the SQ. Combined with the selection rule of the hidden layer, and through experiments, the number of the hidden layer nodes was selected as 11. The final network topology structure is 7-14-1, as shown in Figure 13. The hidden layer of the network adopts the log-sigmoid function, and the output layer adopts the pure line function.

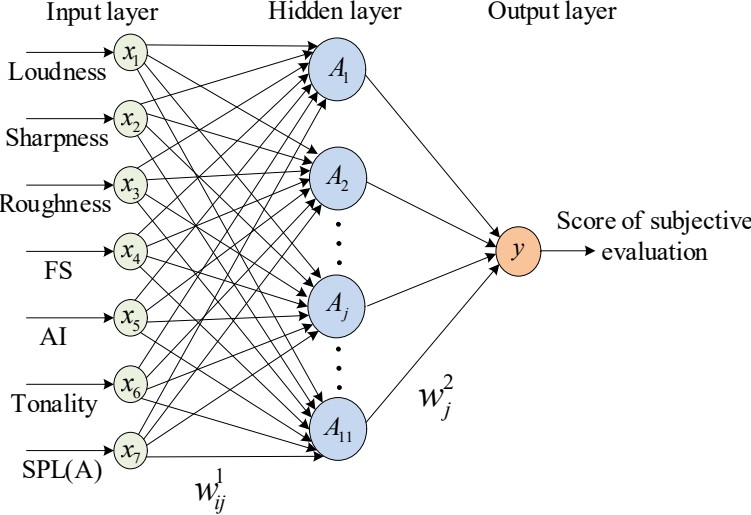

**Figure 13.** Network topology structure of genetic algorithm back propagation artificial neural network (GA-BP ANN).

Before the formal training of the network, the GA algorithm was introduced to optimize the initial weights and thresholds of the BP-ANN. The population size was set to 50, the genetic algebra was 150,

and the mutation probability was 0.05. In this paper, the weights and thresholds were coded using the real number encoding method. After obtaining the individual optimal solution, it was used as the initial weights and thresholds of the BP-ANN.

The subjective and objective evaluation data of the 76 sound samples of the EV were used as training samples, and the remaining 15 sound samples were used as model test samples. The obtained error curve is shown in Figure 14. The training accuracy was revealed to approach 0.001 after 5000 training times, and the important properties of the evaluation model including the convergence speed and the prediction precision were demonstrated and analyzed in the prediction model.

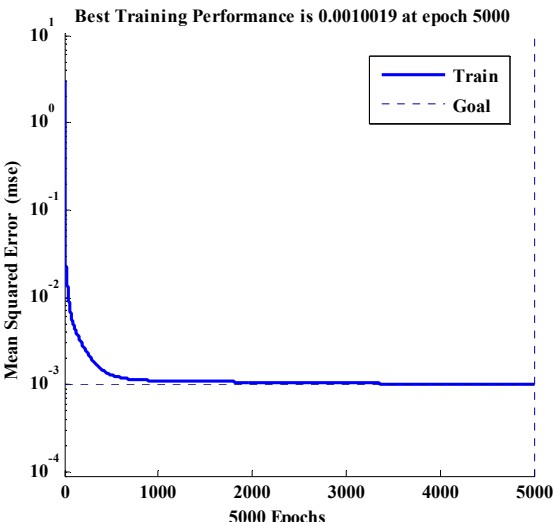

**Figure 14.** The training error curve.

Figure 15 is a scatter diagram of the correlation between the output value of the GA-BP ANN and the actual score, where the correlation coefficient reached 0.928. The result shows that this model can well map objective parameters to the subjective scores of the SQ. Figure 16 shows the relative error between the predicted value of the GA-BP ANN model and the actual subjective score value. The prediction error of the GA-BP ANN model is within 8%, which fully demonstrates the high prediction accuracy of the GA-BP ANN model.

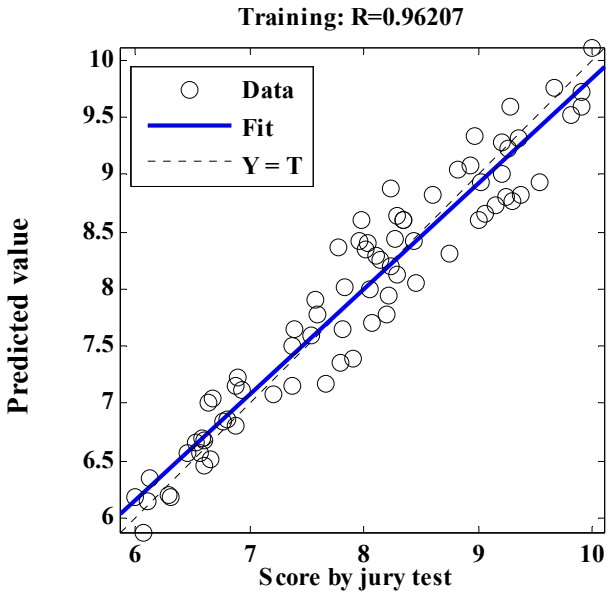

**Figure 15.** Correlation between the prediction value of the model and the scores of the jury test.

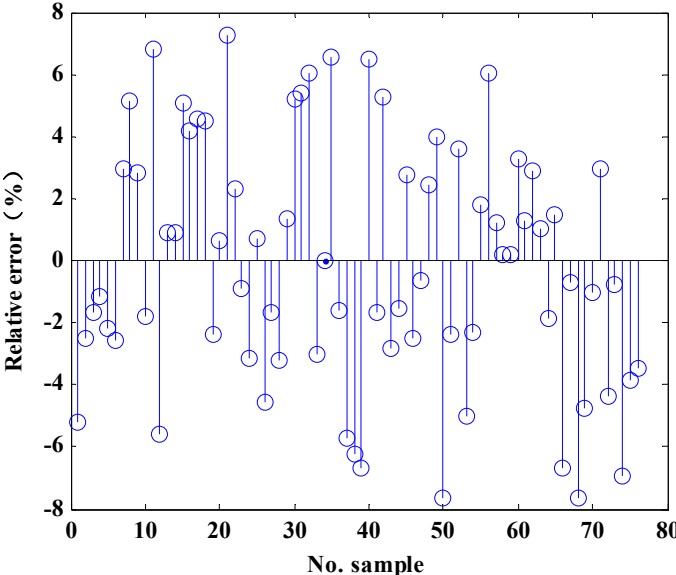

**Figure 16.** The relative error of the GA-BP ANN model.

## 4. Discussion

In order to verify the generalization ability and prediction accuracy of the GA-BP ANN model, the remaining 15 sound samples were used as model verification samples, and the established GA-BP ANN and MLR models were used for the comparative analysis. As shown in Figure 17, the prediction error of the GA-BP ANN model for the SQ in EVs is generally lower than that of the MLR model. Moreover, the average percentage error of the GA-BP ANN model is only 5.81%, which is much lower than the 8.14% of the multiple linear regression model, indicating that the GA-BP ANN model is more accurate in predicting the subjective evaluation score of the SQ.

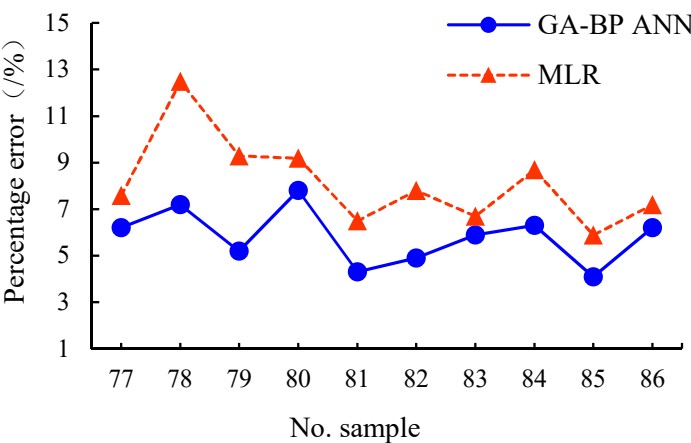

**Figure 17.** Comparison of the percentage error between the actual scores and the predicted values of the two models. MLR, multiple linear regression.

The performance index (PI) is more suitable for comparing the performance of various prediction models. The PI of the evaluation model is evaluated by the distance function $d^2$ between the actual value and the model prediction.

$$d^2 = \frac{1}{n} \sum \left( lg y_i - lg c_i \right)^2, \qquad (2)$$

where $y_i$ is the actual value, $c_i$ is the model prediction value, and n is the number of test samples. The smaller the function $d^2$, the closer the predicted value to the actual value, which reflects the better

predictive performance of the model. Moreover, according to Equation (2), the $d^2$ value of the MLR model is 0.0339, and the $d^2$ value of the GA-BP ANN model is 0.0245, which is far lower than that of the MLR model, indicating that the GA-BP ANN model has better prediction performance. By analyzing the reasons, the subjective sound evaluation for the auditory perception characteristic is a nonlinear process. The established model using the MLR analysis is a linear approximation of the nonlinear process, while the GA-BP ANN model has a good nonlinear input–output relationship and can better simulate the complex nonlinear function relationship, so its accuracy is higher than that of the MLR model. Moreover, the GA-BP ANN model has strong learning and generalization abilities, which are not available in the MLR model.

In the subsequent analysis and evaluation of the SQ of EVs, the psychological objective parameters of sound can be calculated, and the GA-BP ANN evaluation model can be used to predict the subjective scoring values of the sound samples, thus eliminating the tedious and time-consuming subjective hearing tests and simplifying the process. The evaluation process of the GA-BP ANN model compared with the jury test is shown in Figure 18.

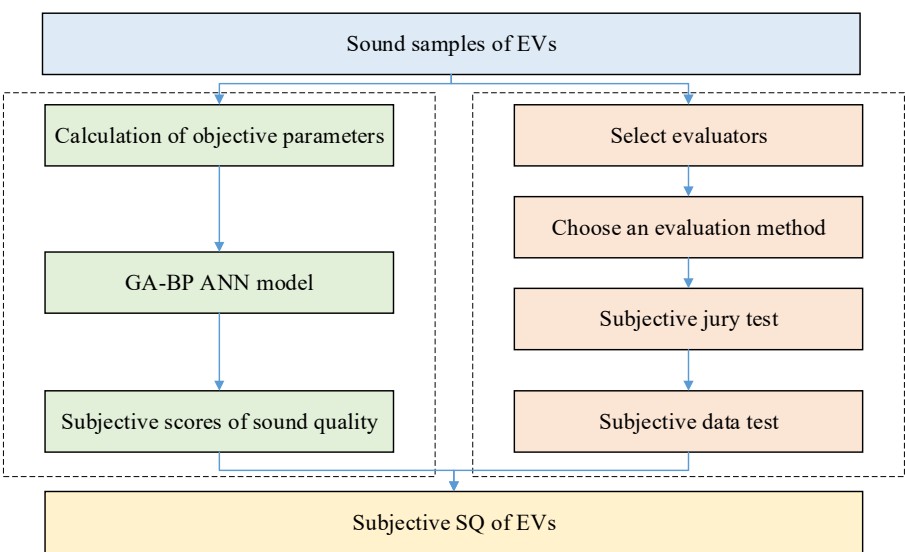

**Figure 18.** Comparison of the evaluation process of the GA-BP ANN model and the jury test.

## 5. Conclusions

In this study, the interior SQ of EVs was successfully evaluated using a GA-BP ANN model. Using the subjective and objective evaluation results, it was possible to develop an SQ evaluation model on the basis of a GA-BP ANN. It was revealed that this model possesses a strong correlation with subjective evaluation and that it also has the ability to transform simplistic objective parameters into subjective perception. Through experimental verifications, it was implied that the GA-BP ANN model possesses excellent generalization abilities. Moreover, compared with the MLR model, the GA-BP ANN model could more accurately and effectively forecast the subjective sensation of the SQ of EVs under the same operating conditions. In addition, the developed model could predict the subjective perception for the SQ of the interior noise generated from EVs only through mapping from the objective physical-psychoacoustical parameters of the noise into the prediction model. Compared with the time-costly process of the subjective jury tests, the model designed in this paper is considerably objective, repetitive, accurate, and effective, and this approach can presumably not only estimate the SQ of EVs, but can also be used as a tool for the SQ improvement process.

**Author Contributions:** Conceptualization, methodology, data curation, funding acquisition, K.Q.; project administration, writing—review and editing, Z.H.; formal analysis, investigation, writing—original draft preparation, D.S. All authors have read and agreed to the published version of the manuscript.

**Funding:** This research was funded by the China Postdoctoral Science Foundation (Grant No. 2019M650657).

**Acknowledgments:** The authors thank all those who participated in the subjective jury test. This work was supported by the China Postdoctoral Science Foundation (Grant No. 2019M650657).

**Conflicts of Interest:** The authors declare no conflict of interest.

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
