# Peer review of "Sound Quality Estimation of Electric Vehicles Based on GA-BP Artificial Neural Networks"

_applsci, doi:10.3390/app10165567_

Round 1
Reviewer 1 Report
The work is properly done but the authors should emphasize innovative elements.
Reviewer 2 Report
Review of manuscript: Sound quality estimation of electric vehicles based on GA-BP artificial neural networks
The manuscript compares two methods for the sound quality evaluation of electric vehicles: multiple linear regression (MLR) analysis and neural networks. Recording of the interior sound were acquired in several vehicles at different speed, and then characterized through a set of objective parameters. A jury test was conducted, and part of the results used to train the neural network. The results indicate that the trained neural network provides results more accurate than the MLR.
The topic is within the scope of the Acoustic and Vibration section of Applied Science.
Major comments
- It is unclear on which basis the subjective evaluation was given. What was the question to which the participants had to answer? What aspect of the sound they were asked to evaluate? Given the selection of a panel of experts, I suppose that they would rate a specific aspect of the sound they heard.
- Why in the regression models SPL(A) was not included?
Minor comments
abstract, line 3: please rephrase. The same SQ evaluation methods can be applied in both cases. Also for ICEVs jury test can be performed with the same drawbacks outlined for EVs.
p.1, line 37: specify the “other acoustic problems” ore remove the sentence.
- 3, line 106: report briefly the recording conditions (e.g. type of road).
- 3, line 115: “there were also 5 seconds after cutting” is unclear.
- 9, line246: replace “SPL(A)” with “FS”.
Reviewer 3 Report
Sound Quality Estimation of Electric Vehicles Based on GA-BP Artificial Neural Networks
The authors present and discuss their work on the use of machine learning as replacement for listening panels in the context of sound quality assessment. The specific subject of this study concerns the auditory perception of the driving noise at the passenger seat in the interior of an electrically driven car.
The work is systematic, complete and well organized. Excellent graphics, formulas and tables.
The results are convincing.
This work is without doubt fit for publication.
Some small details need to be adjusted.
These are the suggestions from this reviewer:
+ 35 makes the subjective perception very irritable better …very irritating
+ 111 ISO 5182 correct to 5128
+ 126 tests were not interfered with by the external environment, better: tests did not interfere with the external environment,
+ 264 The BP-ANN can also easily fall into the local minimum, so it can obtain the global optimal solution. At first sight this seems contradictory…
+ 273 afterward correct to afterwards
Round 2
Reviewer 2 Report
The Authors address my concerns in a satisfactory manner.